# Spectroscopic signature of obstructed surface states in SrIn$_2$P$_2$

Xiang-Rui Liu [1,2,6], Hanbin Deng [1,2,6], Yuntian Liu [1,2,6], Zhouyi Yin[1], Congrun Chen[1], Yu-Peng Zhu [1,2], Yichen Yang[3], Zhicheng Jiang[3], Zhengtai Liu [3], Mao Ye[3], Dawei Shen [3], Jia-Xin Yin[4], Kedong Wang [1], Qihang Liu [1,2,5] ✉, Yue Zhao [1,2] ✉ & Chang Liu [1,2] ✉

The century-long development of surface sciences has witnessed the discoveries of a variety of quantum states. In the recently proposed "obstructed atomic insulators", symmetric charges are pinned at virtual sites where no real atoms reside. The cleavage through these sites could lead to a set of obstructed surface states with partial electronic occupation. Here, utilizing scanning tunneling microscopy, angle-resolved photoemission spectroscopy and first-principles calculations, we observe spectroscopic signature of obstructed surface states in SrIn$_2$P$_2$. We find that a pair of surface states that are originated from the pristine obstructed surface states split in energy by a unique surface reconstruction. The upper branch is marked with a striking differential conductance peak followed by negative differential conductance, signaling its localized nature, while the lower branch is found to be highly dispersive. This pair of surface states is in consistency with our calculational results. Our finding not only demonstrates a surface quantum state induced by a new type of bulk-boundary correspondence, but also provides a platform for exploring efficient catalysts and related surface engineering.

Surface states of truncated solids, whose wave functions are spatially confined at the boundaries and exponentially decay into the bulk, give rise to a low-dimensional dispersion distinct from that of the bulk state. Typically, various physical and chemical environments of the atomic sites on the surface determine the electronic nature of the surface states. In quantum materials, surfaces and interfaces provide a fertile playground for emergent phenomena at lower dimensions[1–4], which include but are not limited to the two-dimensional electron gas[5], the Rashba spin-split surface states on terminations of noble metals[6,7], and the topological surface states of topological insulators (TIs)[8] and topological semimetals[9,10]. In topological materials, surface states are more exotic in a sense that they also reflect the topological

properties inside the bulk via the nontrivial bulk-boundary correspondence.

It is widely accepted that when a surface termination is formed in an ordinary insulator, no matter topologically trivial or nontrivial, the atoms on the topmost layer are either completely peeled off or left intact at the termination (Fig. 1a). However, there are also insulators that the ground-state charge centers locate at some "virtual sites" where no atoms reside. The cleavage through these virtual sites would lead to fractional occupied charges (also dubbed "filling anomaly", Fig. 1b)[11–14] which is in close analogy to a three-dimensional (3D) version of the Su-Schrieffer-Heeger model[15,16] that carries half an electric charge at a domain wall, giving rise to "obstructed surface states" that

[1]Shenzhen Institute for Quantum Science and Engineering (SIQSE) and Department of Physics, Southern University of Science and Technology (SUSTech), 518055 Shenzhen, Guangdong, China. [2]International Quantum Academy, 518048 Shenzhen, Guangdong, China. [3]State Key Laboratory of Functional Materials for Informatics, Shanghai Institute of Microsystem and Information Technology, Chinese Academy of Sciences, 200050 Shanghai, China. [4]Department of Physics, Princeton University, Princeton, New Jersey 08544, USA. [5]Guangdong Provincial Key Laboratory of Computational Science and Material Design, Southern University of Science and Technology, 518055 Shenzhen, Guangdong, China. [6]These authors contributed equally: Xiang-Rui Liu, Hanbin Deng, Yuntian Liu. ✉e-mail: liuqh@sustech.edu.cn; zhaoy@sustech.edu.cn; liuc@sustech.edu.cn

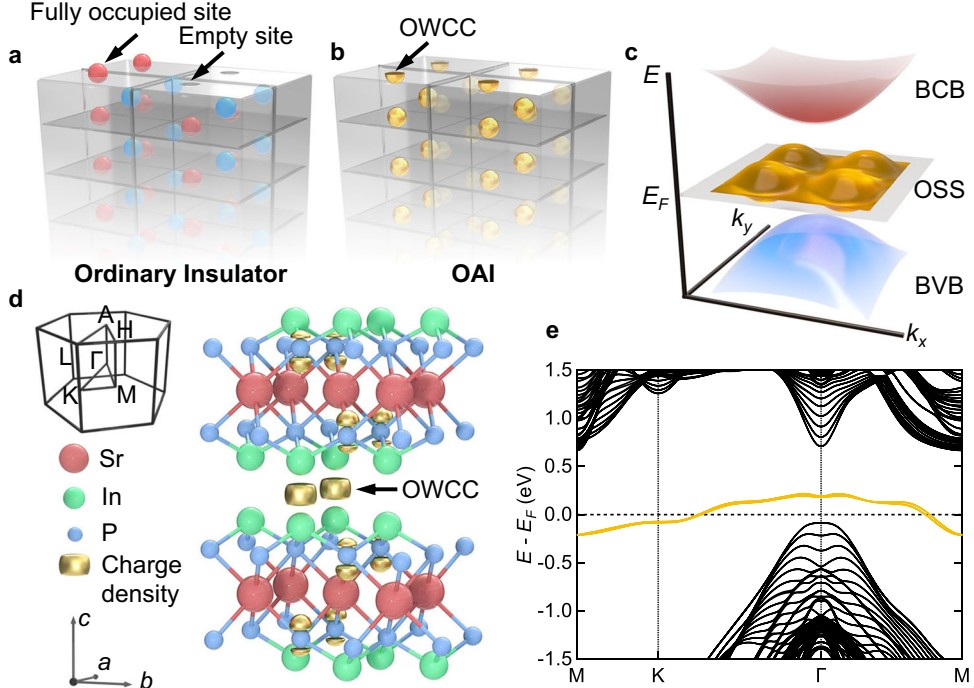

**Fig. 1 | Concepts of obstructed atomic insulator (OAI) and appearance of OAI phase in SrIn₂P₂.** **a**, **b** Schematic atoms and Wannier charge centers (WCCs) near the cleavage plane of (**a**) an ordinary insulator and (**b**) an OAI. Red and blue balls in **a** represent the indivisible atoms. Golden balls and surface hemispheres in **b** represent the obstructed Wannier charge centers (OWCCs) located away from the atoms. Gray blocks represent the unit cells with a top surface. For ordinary insulators, some of the atoms were left intact on the interface while others were gone totally. For OAI, to preserve the symmetry and charge neutrality of the system simultaneously, partial charges (golden hemispheres) were left on the interface.

**c** Illustration of the metallic obstructed surface states (OSSs) that originate from the OWCCs at the interface within the bulk band gap. BCB: bulk conduction band, BVB: bulk valence band. **d** Schematic crystal structure and Brillouin zone of SrIn₂P₂, overlaid with calculated charge distribution (golden shapes). Charges were found to accumulate on the OWCCs that lie in the quintuple layer gaps. **e** DFT-calculated bulk band structure of the In-terminated (0001) cleavage plane of SrIn₂P₂ with a 7-unit-cell slab model, in which the Γ and M points are 4-fold degenerate if the hybridization between the two surfaces is ignored. The surface projected bands are represented by the orange lines.

---

are pinned to cross the Fermi level due to their partial filling (Fig. 1c). Recently, such unconventional insulators, named the obstructed atomic insulators[17–23], are found to include various materials such as electrides[24–27], high-order TIs and hydrogen evolution reaction electrocatalysts[21,23]. The charge centers localized at such virtual sites, named the obstructed Wannier charge centers, would in principle be active for ligand adsorption and electron transfer, making obstructed atomic insulator an ideal platform for surface catalysis and related applications[21,23,28–30].

In realistic electronic systems, it is challenging to observe the fractionalization of charges at the termination, as surface structural/electronic reconstruction may tend to pair up the fragmented electrons and remove the filling anomaly. However, such reconstruction will not change the presence of obstructed Wannier charge centers inside the bulk. As a result, an unforeseen type of surface state would inevitably be left on the surface according to bulk-boundary correspondence. In this work, we utilize scanning tunneling microscopy and spectroscopy (STM and STS), angle-resolved photoemission spectroscopy (ARPES)[31,32], and density functional theory (DFT) calculations to realize this new class of surface state on an obstructed atomic insulator candidate material SrIn₂P₂. Our STM topography reveals a $\sqrt{3} \times 1$ structural reconstruction of terminated In atoms on its (0001) cleavage plane, giving rise to a stripe-like pattern of the charge density. In such reconstructed areas, an unusual, highly-localized in-gap state with relatively narrow bandwidth above the Fermi level is observed, signified by the presence of a persistent negative differential conductance following a strong STS peak all over the defect-free area on the crystal surface. ARPES data resolves another surface state situating largely below the Fermi level, as well as the bulk valence bands of the

system with strong $k_z$ dispersion. DFT calculations show that while the undistorted SrIn₂P₂ is an obstructed atomic insulator, its surface tends to form $\sqrt{3} \times 1$ reconstruction, which dimerizes the half charges of the parent obstructed surface states. In this situation, the original, filling-anomaly-induced obstructed surface states split in energy, giving rise to the ARPES-observed surface state (lower branch) and the STS-resolved flat, localized state followed by unusual negative differential conductance above the Fermi level (upper branch). These new bands comprise a novel form of bulk-surface correspondent electronic states. Our finding paves the way for further research on obstructed atomic insulators and enriches the connotation of surface electronic structure.

## Results
### Predicted obstructed surface states and presence of surface reconstruction

SrIn₂P₂ has a hexagonal crystal structure with quintuple layers (QLs) stacking along the c axis, rendering a space group P6₃/mmc with lattice constants $a = b = 4.0945$ Å and $c = 17.812$ Å[33]. Sr atoms occupy the Wyckoff position 2a while both In and P atoms occupy the Wyckoff position 4f, as shown in Fig. 1c. Using the real space invariant indices[18,21], Xu et al. show that SrIn₂P₂ and its isostructural partner CaIn₂P₂ possess obstructed Wannier charge centers at Wyckoff positions 2d, where no atoms are present. Our DFT-calculated bulk charge density distribution (Fig. 1d) shows that the interlayer charges mainly locate at such empty sites, indicating that the AIn₂P₂ (A = Ca, Sr) compounds are obstructed atomic insulators whose (0001) cleavage plane between two In atoms cuts through the obstructed Wannier charge centers without intersecting with any real atoms. The calculated band structure of a 7-unit-

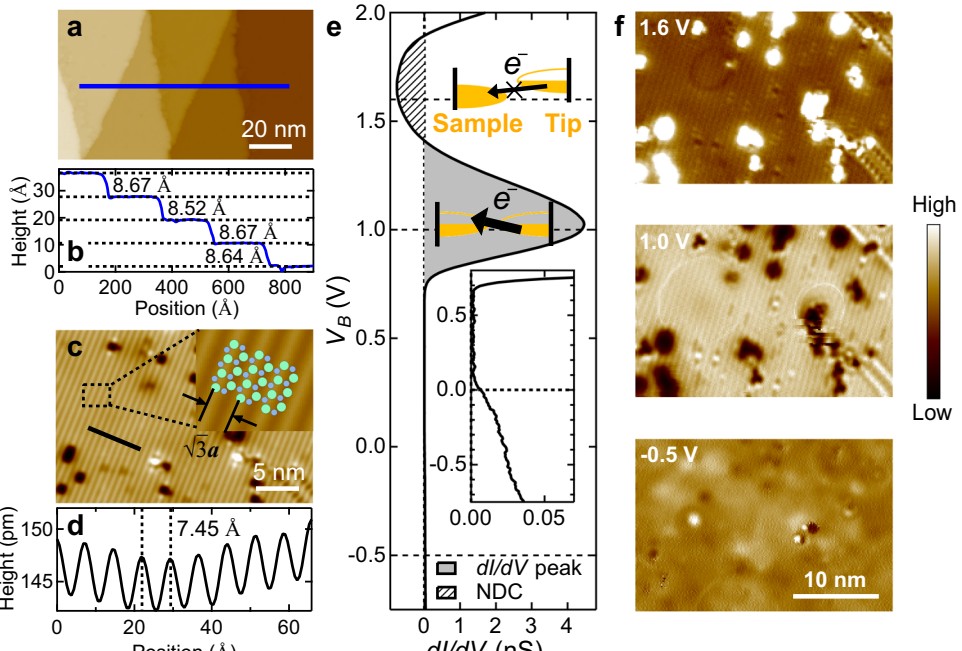

**Fig. 2 | Structural reconstruction and the strongly-localized in-gap state on the SrIn$_2$P$_2$ (0001) cleavage plane. a** STM topographic image on a freshly cleaved surface. **b** The height profile along the blue line, revealing a step height of -8.6 Å between two neighboring terraces – the half height of a unit cell. **c** STM topographic image zoomed in on a single terrace, showing a stripe-like pattern. Inset shows an enlarged area overlaid with the surface atomic structure. Green/blue balls: In/P. **d** Height profile along the black line, showing a modulation with a period of 7.45 Å ($\sqrt{3}a$), which indicates a $\sqrt{3} \times 1$ structural/electrical reconstruction. **e** STS spectra taken on the $\sqrt{3} \times 1$ reconstructed region, showing a strong peak that centers at bias voltage $V_B = 1.0$ V followed by a region of negative differential

conductance (NDC) at higher biases. The gray region indicates the d$I$/d$V$ peak, while the stripe-shaded region indicates the NDC. Schematic insets describe the resonant tunneling between the tip and the sample surface that yields the spectral peak and the NDC. Data inset shows the STS zoomed in from −0.75 V to 0.8 V for observation of non-zero LDOS at $V_B <$ 100 mV. **f** Selected d$I$/d$V$ maps taken at corresponding biases on the same spatial region as in **c**, showing typical real-space electronic structures in negative bias (−0.5 V), peak region (1.0 V), and NDC region (1.6 V). Tunneling parameters: **a, b** $V_B = 1.6$ V, $I_t = 100$ pA; **c, d** $V_B = 1.6$ V, $I_t = 500$ pA; **f** $V_B = 1.6$ V, $I_t = 500$ pA for the top two panels, and $V_B = −0.7$ V, $I_t = 500$ pA for the bottom panel.

cell slab of undistorted SrIn$_2$P$_2$ is shown in Fig. 1e. We find that the relatively-flat surface states with Kramers degeneracy at time-reversal invariant momenta (Γ and M) are located inside the direct band gap (-0.8 eV). Half occupation of these states (filling anomaly) inevitably occurs if charge neutrality and certain crystalline symmetries (e.g., inversion symmetry of the slab) are simultaneously preserved[11–14], giving rise to metallic obstructed surface states.

Single crystals of SrIn$_2$P$_2$ were grown using the flux method described in ref. 33 and characterized by single crystal x-ray diffraction and in-situ core level photoemission measurements (Fig. S1). Both results indicate the high quality of our single crystals. As shown in Fig. 2a, b, the STM topography image on the cleaved surface measures a step height of 8.7 Å between two neighboring terraces, which approximates the half height of a unit cell. This step height, combined with the smallest energy found for the $c$-oriented In-In bonds between adjacent QLs (Table S1), provides solid evidence that the cleavage happens on the (0001) plane between two In atoms of two adjacent QLs – where obstructed surface states should be present if no surface reconstruction appeared.

Yet, the topography of the cleavage plane at $V_B = 1.6$ V reveals an ordered stripe-like pattern instead of the expected ideal In lattice (Fig. 2c). The height profile along the direction perpendicular to the stripes shows a modulation with a period of $\sqrt{3}a$, indicating a $\sqrt{3} \times 1$ structural reconstruction of such cleaved surface (Fig. 2d) where one of the In atoms in two consecutive in-plane unit cells is levitated away from the surface. Domains of several micrometers with stripes along all three directions with an angle of 120° can be observed. Occasionally, stripes with $2a$ spacing are observed with <5% coverage in our scans (Fig. S2).

### Spatially localized surface state above $E_F$

To investigate the presence of the filling-anomaly-induced obstructed surface states under surface reconstruction, we perform a systematic spectroscopic study on a well-ordered $\sqrt{3} \times 1$ reconstructed region (See also Fig. S5). The differential conductance d$I$/d$V$ (proportional to the local density of states (LDOS)) shows a dramatic peak near 1.0 V followed by a region with negative differential conductance (Fig. 2e). The energy location of this STS peak is found to be sample dependent, as we have also observed the peaks located at 0.7 V on another spatial region of the same sample (Fig. S7). A global gaplike flat bottom is observed from −0.5 V until the peak appears. A close-up look at the STS spectra reveals small but finite LDOS below $V_B \sim 0.1$ V (Inset of Fig. 2e), which is not only in agreement with the semiconducting nature of SrIn$_2$P$_2$, but also consistent with the observation of bands below $E_F$ in the ARPES measurements (Fig. 3).

Importantly, the spectral peak, followed by negative differential conductance, rises up rapidly from the gap at around 0.7 V and its intensity is so strong that the STS signals near the Fermi level are rendered barely visible (Fig. 2e). As shown in the insets of Fig. 2e, the apex of the STM tip is spatially localized. When such a spatially-localized electronic state sweeps through the Fermi level of the sample surface with increasing bias voltages, the rapid filling of the surface state results in a resonant peak and the associated negative differential conductance. It should be highlighted here that the observation of negative differential conductance following a strong STS peak in a single crystal surface is particularly rare, indicating both a high LDOS and a resonant tunneling structure near the high LDOS energy level. In previous STS measurements, this phenomenon is mostly associated with spatially localized, narrow-band-width defects[34–36] and chemical

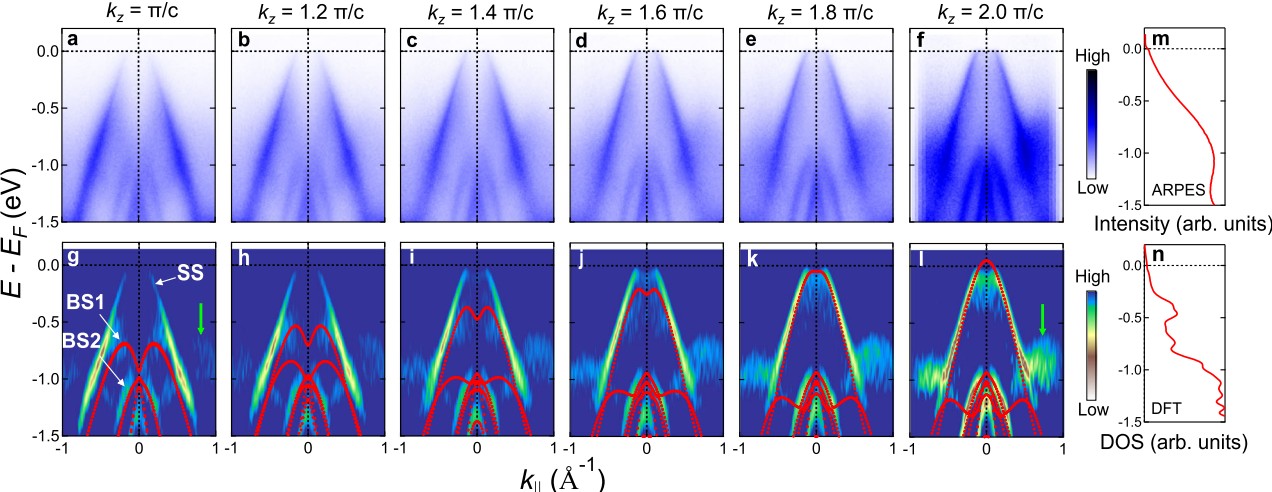

**Fig. 3 | Surface and bulk electronic structure of SrIn$_2$P$_2$, showing the OSS-derived surface bands below the Fermi level. a–f** ARPES $E$-$k$ cuts along $\bar{K}$-$\bar{\Gamma}$-$\bar{K}$ at different $k_z$s. **g–l** Second order curvature analysis along the energy distribution curves (EDCs) for the corresponding spectra in **a–f**. Red dots: DFT-calculated bulk bands after an energy upshift of 0.25 eV; white arrows: dispersion of the OSS-derived surface state (SS) and the bulk bands BS1 and BS2; green arrows: tail of the SS outside the hole-pocket region. **m** ARPES partial density of states (PDOS) obtained by integrating the ARPES intensity along the $k_{||}$ axis from **a** to **f**. **n** DFT-derived density of states (DOS) at the same energy region as the ARPES maps. Both ARPES PDOS and DFT DOS increase with increasing binding energy, while approaching zero towards the Fermi level. The consistency between our ARPES-derived PDOS, d$I$/d$V$ curve (inset in Fig. 2e) and DFT-derived DOS indicates an In-terminated (0001) cleavage plane where both the SS and BS hole bands are expected to top slightly above $E_F$ at $\bar{\Gamma}$.

changes in the reaction processes[37]. In our case, however, both the peak and the negative differential conductance are persistent on well-ordered regions spanning the whole pristine surface except for the defects (Fig. S5 and S6). This implies that such electronic states are intrinsic on the sample surface, suggesting a novel type of surface electronic structure. This phenomenon also elucidates the key difference between ordinary surface states with high charge density and the surface states induced by obstructed Wannier charge centers. (The mechanism of negative differential conductance and its relation to the surface state is discussed in detail in section IX of the Supporting Information).

We then perform detailed d$I$/d$V$ mapping on the same region from $V_B = -0.7$ to 1.6 V to study the spatial distribution of this novel electronic structure (Fig. 2f and Figs. S8, S9). The d$I$/d$V$ maps show a much stronger signal within $V_B = 0.9$ to 1.2 V, which agrees with the pronounced peak seen on the STS spectra. The rapid filling of the observed surface state within these biases likely leads to the contrast inversion at the defect sites in the d$I$/d$V$ maps. It is worth noting that a stripe-like pattern similar to the topographic image is clearly observed at the peak region, while no such pattern was observed at $V_B = -0.5$ V below the Fermi level, away from the gap (Fig. 2f). Such spatial distribution of LDOS, characterized by a punctiform array of localized in-plane charge centers on the terminated surface, is evident in its relation to the new electronic state, which is detailed in Fig. 4.

### Highly dispersive surface state below $E_F$

To probe the $k$-dependent electronic structure on the reconstructed termination of SrIn$_2$P$_2$ below the Fermi level, high-resolution ARPES measurements were performed. Fig. 3a–f show the overall band structure of the SrIn$_2$P$_2$ (0001) cleavage plane along $\bar{\Gamma}$-$\bar{K}$ at different $k_z$s ($\pi/c < k_z < 2\pi/c$, corresponding to $90 < h\nu < 100$ eV at $\bar{\Gamma}$); Fig. 3g–l show the same spectra after applying a second-order curvature analysis[38] along the energy distribution curves, overlaid with the corresponding DFT calculation results of the bulk states with an energy upshift of 0.25 eV (red dots), which accounts for possible intrinsic hole doping of the sample. From $k_z = \pi/c$ to $k_z = 2\pi/c$, BS1 evolves from a M-shaped band with band top at binding energy $E_b = 0.6$ eV to a hole pocket likely crossing $E_F$, showing steep $k_z$ dispersion along $\Gamma$-A; BS2 changes its

shape while shifting slightly down in energy, yielding a relatively mild $k_z$ dispersion. The constant energy contours within $E_b = 1.0$ eV show isotropic, circular shapes of these bands, while for $E_b > 1.0$ eV the bands show clear six-fold symmetry (Section X and Figs. S12–S13 in the Supporting Information). The low signal-noise ratio of these bulk bands may come from the $k_z$ broadening effect[39,40] or the matrix element effect[32]. Overall, the excellent agreement between the experiments and the DFT results as well as the substantial $k_z$ dispersion of these states verifies their bulk nature.

Importantly, besides the bulk bands, two bands dispersing from the Fermi level to binding energies higher than 1.5 eV can be observed in all spectra (labeled "SS" in Fig. 3g–l). Additional ARPES intensity was also seen outside the hole pocket at each $k_z$ (green arrows in Fig. 3g, l), which cannot be attributed to any bulk states. In fact, residue intensity of this state extends to both $\bar{K}$ and $\bar{M}$ (Figs. S15 and S16), the dispersion of which is consistent with the DFT result for the lower branch of surface states originating from the obstructed surface states. Systematic photon-energy-dependent maps were further performed in a wide range of photon energies (Section XI and Fig. S14 of the Supporting Information). From our data, these bands have no $k_z$ dispersion over half of the out-of-plane Brillouin zone, signifying their two-dimensional, surface-like nature.

Finally, we integrate the ARPES intensities along the $k_{||}$ axes of Fig. 3a–f and adding up the resultant $I$ vs. $E$ curves (Fig. 3m). This method gives a reasonable estimation of the ARPES-observed partial density of states, as the data in Fig. 3a–f covers essentially a half of the 3D Brillouin zone. Comparing the ARPES partial density of states with the DFT-derived DOS (Fig. 3n) and the STS spectra that focus on low bias voltages (Inset of Fig. 2e), we judge that both the ARPES-resolved bulk and surface states extend in energy up to ~100 meV above $E_F$, where the DFT DOS and the d$I$/d$V$ curve begin to show nonzero values. This comparison justifies the consistency between our STM and ARPES measurements as well as our DFT calculations.

### Adiabatic evolution of the obstructed surface states

Till now, we have gathered the following information about the new surface states observed in SrIn$_2$P$_2$: (i) According to the theory of obstructed atomic insulators, a pair of relatively flat, metallic

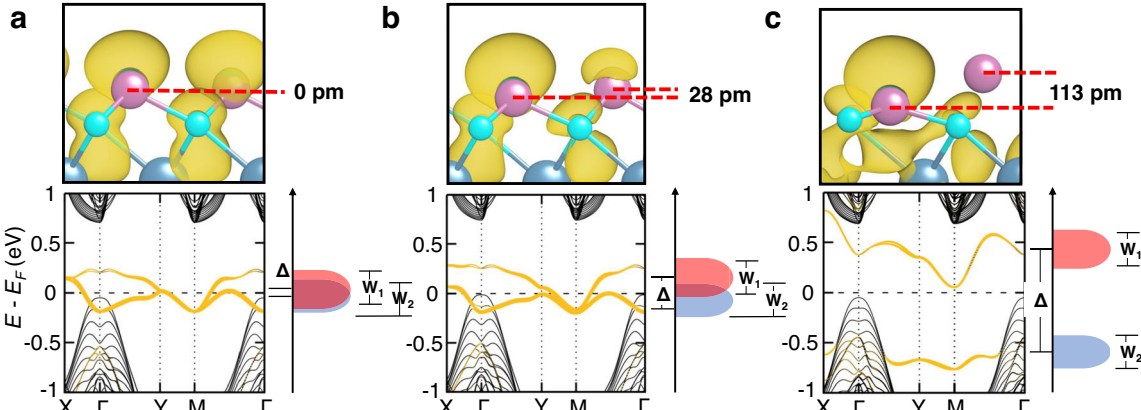

**Fig. 4 | DFT calculation on the $\sqrt{3}\times 1$ reconstructed surfaces with different vertical shifts of the In atoms.** The relative shift of the In atoms is **a** fixed to be zero, **b** fixed to be 28 pm along *c*, and **c** 113 pm along *c* (fully relaxed). Orange bands in the lower panels represent the surface projected states. Golden regions in the top panels represent the charge distributions of the upper branch of the split SS. Δ: average energy difference of the two branches of SS; $W_1$ and $W_2$: bandwidths of the two branches. Δ is found to increase as the shift of In atoms increases, while $W_1$ and $W_2$ are essentially unchanged. As a result, the two branches gradually split in energy, and the charge distribution becomes increasingly unbalanced. For the fully relaxed situation (**c**), the band structure becomes fully gapped, and the charges on the topmost layer accumulate around only one of the two In atoms.

obstructed surface states should have appeared due to the filling anomaly for the undistorted surface. However, such half-filled bands are not observed both in our STM and ARPES measurements. (ii) Our STM topographic maps resolved clear $\sqrt{3}\times 1$ structural reconstruction that covers most of the as-cleaved (0001) surface, with a significant height difference between neighboring In atoms on the surface. (iii) Our ARPES and STS data clearly resolve two sets of surface states, one situates above $E_F$ with relatively narrow bandwidth, showing unusual spatial distribution; the other locates mostly below $E_F$, with a broader bandwidth of ~1.5 eV.

To reveal the relationship between these surface states and the obstructed surface states of the undistorted case, we perform DFT calculations of the surface electronic structure (see Methods). First, we choose a 7-unit-cell slab and apply full relaxation to examine whether the experimentally observed $\sqrt{3}\times 1$ reconstruction is energetically favorable. Indeed, our calculations show that the two In atoms of the outermost layer in the $\sqrt{3}\times 1$ supercell are 113 pm elevated along the *c* axis after relaxation (Fig. 4c), while the positions of other atoms are almost unchanged. Such $\sqrt{3}\times 1$ reconstruction lowers the surface energy by 26 meV/Å² compared with the primitive surface cell (Section XIII and Fig. S17 in the Supporting Information), well explaining the observed spontaneous reconstruction upon cleavage.

Furthermore, our calculations find that the two sets of surface states observed by our STM and ARPES measurements are indeed splitting obstructed surface states due to the surface reconstruction. To understand such a mechanism more clearly, we show different surface state calculations in the $\sqrt{3}\times 1$ supercell where the displacement between two neighboring In atoms are consecutively increasing (Fig. 4a–c). For the undistorted case (Fig. 4a), the two branches of obstructed surface states float at the Fermi level; filling anomaly and the fractional occupation of charges are present because of the fourfold degeneracy at the Y and M points owing to the band folding effect. When the displacement between In atoms is manually set to 28 pm (Fig. 4b), the in-plane translational symmetry is broken. Such symmetry breaking means that the In atoms at the surface are no longer occupying the Wyckoff positions with obstructed Wannier charge centers in the bulk. As a result, the energy degeneracy at the Y and M points is lifted, violating the filling anomaly. However, as long as the bandwidths of the upper and lower branches ($W_1$ and $W_2$) are larger than the reconstruction-induced band splitting Δ, an overall bandgap is still absent. In this case, the fractional occupation is still present, with

the two neighboring In atoms carrying unbalanced charge densities. Finally, when the neighboring In atoms are fully relaxed to 113 pm apart vertically (Fig. 4c), Δ overwhelms $W_1$ and $W_2$, leading to an overall gap between the two obstructed surface state branches and a further enhanced imbalance of charge density for the two branches, which is exactly what we have observed experimentally. The upper branch is consistent with the pronounced STS peak, though the measured width of the peak is smaller than the theoretical bandwidth, which is possibly due to the strong tunneling sensitivity to the states near the $\bar{\Gamma}$-point. The lower branch agrees with the two-dimensional band observed in ARPES, both in its hole-like nature near $\bar{\Gamma}$ and its tails that flutter away from $\bar{\Gamma}$ (green arrows in Figs. 3g, l and S16c, d).

Therefore, in obstructed atomic insulator candidates, the spatially localized fractional charge left on the empty sites of the surface could lead to unstable metallic states. In response, energy minimization tends to break the fragile situation by surface reconstruction, as our case exemplifies. However, the nature of the bulk-obstructed Wannier charge centers, as well as the bulk-boundary correspondence, remains intact. In this sense, we could treat such reconstruction as an "adiabatic" evolution from the obstructed atomic insulator case, while the bulk-boundary correspondence demonstrates the robustness of such in-gap surface states even without filling anomaly on the reconstructed surface. The localized feature of such surface state is not affected by the reconstruction, as verified by the robust negative differential conductance observed in our STS measurements.

## Discussion

As a concluding remark, here we compare the obstructed surface states demonstrated in this work to other well-studied surface electronic structures. On one hand, the traditional Tamm and Shockley surface states describe soliton solutions to the Schrödinger equation that are confined to the surfaces of a solid-state system, exhibiting no logical link with the bulk electronic structure. On the other hand, the celebrated topological surface states are constructed out of the bulk electronic wave functions, exemplifying nontrivial bulk-boundary correspondence. The obstructed surface states studied here in a sense bridges the two: the real-space appearance of this surface state resembles that in materials with surface dangling bonds, while its occurrence originates from the bulk electronic structure where the presence of charge centers on empty sites between real atoms is protected by topological invariants. Compared with the surface states

created by ordinary dangling bonds in covalent compounds or by the STM tip where localized states only reside at some discrete sites[41–43], here the strongly-localized state is persistent over a well-ordered cleavage surface, providing huge amount of active sites for atoms/molecules adsorption and electron transfer. These features make the obstructed atomic insulators promising candidates for photocatalysis with high efficiency and easy access.

In summary, we perform systematic STM and ARPES measurements on the (0001) cleavage plane of $SrIn_2P_2$. These spectroscopic experiments resolve an extraordinary localized surface state with rarely reported negative differential conductance above the Fermi level, and a two-dimensional surface state below $E_F$. Our DFT calculations with reconstructed surface termination well reproduce the band dispersion observed by ARPES and agree with the localized surface state observed by STS, revealing the evolution of the obstructed surface states upon surface reconstruction. Our finding demonstrates signatures of the reconstructed obstructed surface states on the (0001) cleavage plane of $SrIn_2P_2$, which sheds light on the research of obstructed atomic insulators and provides a platform for exploring efficient catalysts based on a new mechanism.

## Methods

### Sample growth
Single crystals of $SrIn_2P_2$ were grown using the self-flux method[33]. Small pieces of Sr (Alfa Aesar, 99%), blocks of In (99.9%), and powders of P (Alfa Aesar, 99%) was mixed with a molar ratio Sr: P: In = 3: 6: 110 in an alumina crucible, and flame-sealed in a quartz ampoule under Argon protection. The sealed ampoule is set in a box furnace. The furnace is heated from room temperature to 1373 K in 20 h, then maintain at this temperature for 16 h and slowly cool down to 1146 K in 120 h. 24 h later, the ampoule is centrifugalized to remove the In flux. Shiny, millimeter-sized crystals of $SrIn_2P_2$ can be obtained at the bottom of the crucible. Single crystal x-ray diffraction was performed with Cu K$\alpha$ radiation at room temperature using a Rigaku Miniex diffractometer.

### STM and STS measurements
The STM results are obtained with a Unisoku USM-1500 system equipped with a low-temperature cleaving stage. The $SrIn_2P_2$ sample is cleaved at ~ 7 K with the cooling of liquid helium in ultra-high vacuum (better than $2 \times 10^{-10}$ Torr). The cleaved sample is then immediately transferred to the STM module. All the STM measurements in this paper were done at 5 K (measured with a Diode temperature sensor on the stage). The topographic images are measured in the constant current mode with a tungsten tip pre-treated by an electron-beam heater (Model EBT-100). Standard lock-in method is used in the scanning tunneling spectroscopy (STS) measurements. The frequency of the modulation voltage is 971.9 Hz.

### ARPES measurements
High-resolution ARPES measurements were performed at Beamline 03U of the Shanghai Synchrotron Radiation Facility equipped with a Scienta DA30 electron analyzer[44,45]. The energy and angular resolution were set to be better than 20 meV and 0.2°, respectively. The sample is cleaved in-situ by the top-post method at 15 K after transferring into the chamber. During the measurement, the temperature of the sample was kept at ~15 K, and the pressure was better than $7 \times 10^{-11}$ mbar. The incident beam is $p$-polarized.

### First-principles calculations
The first-principles calculations were carried out by the Vienna ab initio simulation package (VASP)[46] based on the projector augmented wave (PAW) method[47]. The exchange-correlation functional was described by the generalized gradient approximation with the Perdew-Burke-Ernzerhof formalism (PBE)[48]. The plane-wave cutoff energy was set to 350 eV and the van der Waals correction was included via the DFT-D3

method[49]. To study the surface of the crystal, we constructed a slab structure with the thickness of 7unit cells. The in-plane lattice constant is set to $a = 4.0945$ Å[33]. The whole Brillouin-zone was sampled by a $9 \times 9 \times 1$ Monkhorst-Pack grid for the undistorted slab. The surface reconstruction of the crystal was captured by a $\sqrt{3} \times 1$ supercell in the $ab$ plane with a $7 \times 9 \times 1$ Monkhorst-Pack grid. For the results in Fig. 4c, all the atoms were fully relaxed until the force on each atom was less than 0.001 eV/Å. The total energy convergence criteria was set to $1.0 \times 10^{-7}$ eV.

## Data availability
Data are available from the corresponding authors upon reasonable request.

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

## Acknowledgements

Work at SUSTech was supported by the National Key R&D Program of China (Grant Nos. 2022YFA1403700 and 2020YFA0308900), the National Natural Science Foundation of China (NSFC) (Nos. 12074161, 11504159, 11674150), NSFC Guangdong (No. 2016A030313650), the Key-Area Research and Development Program of Guangdong Province (2019B010931001), Guangdong Provincial Key Laboratory for Computational Science and Material Design (No. 2019B030301001) and the Guangdong Innovative and Entrepreneurial Research Team Program (Nos. 2016ZT06D348, 2017ZT07C062). The ARPES experiments were performed at BL03U of Shanghai Synchrotron Radiation Facility under the approval of the Proposal Assessing Committee of SiP.ME$^2$ platform project (Proposal No. 11227902) supported by NSFC. The DFT calculations were performed at Center for Computational Science and Engineering of Southern University of Science and Technology. D.S. acknowledges support from NSFC (No. U2032208). Y.Z. acknowledges support from the Shenzhen High-level Special Fund (No. G02206304, G02206404). C.L. acknowledges support from the Highlight Project (No. PHYS-HL-2020-1) of the College of Science, SUSTech.

## Author contributions

X.-R.L. and C.L. conceived and designed the research project. X.-R.L. grew and characterized the single crystals. H.D., Z.Y., C.C., J.-X.Y., K.W., and Y.Z. performed the STM measurements. X.-R.L., Y.-P.Z., Y.Y., Z.J., Z.L., M.Y., D.S., and C.L. performed the ARPES measurements. Y.L. and Q.L. performed the DFT calculations. X.-R.L., H.D., Y.L., Y.Z., Q.L., and C.L. wrote the paper.

## Competing interests

The authors declare no competing interests.
