## [Peer Review File · Nature Communications]

Spectroscopic signature of obstructed surface states in SrIn₂P₂REVIEWER COMMENTS

Reviewer #1 (Remarks to the Author):

In the present manuscript, Liu et al. utilize STM and ARPES measurements as well as DFT calculations to study the low-lying electronic structure of SrIn₂P₂, demonstrating a new type of surface state that originated from the so-called “obstructed surface states (OSS)”. Such surface states appear on obstructed atomic insulators, a novel class of materials distinguished from trivial atomic insulators with potential applicability on surface catalysis. The observed surface states differ from the theoretical OSSs in that they split in energy due to a surface structural reconstruction, rendering two branches of surface bands, one on each side of the Fermi level. This reconstruction eliminates the partial occupancy of electrons in the original OSS picture, yet preserves the spatial locality of the OSSs, which is key ingredient for catalytic activity.

I found that this manuscript provided solid experimental and computational evidence supporting the observation of this surface state. To my knowledge, it is the first report of spectroscopic realization of such previously unstudied low-dimensional electronic structure. The STM and ARPES data is systematic and of high quality, sufficient to rule out other possibilities of the observed phenomena. The DFT calculations also provided a rational scenario to explain the experimental observations. Based on these points, I tend to recommend this paper to be published in Nature Communications, provided the following comments are addressed appropriately.

1. I wonder the difference between the surface band structure of OWCC termination and other terminations that can exist in SrIn₂P₂ (such as breaking Sr-P or In-P bond). For the OAI system, will they carry special surface states when the termination does not cross the OWCC?
2. For the surface reconstruction supported by DFT calculation, the stripes along three directions with an angle of 120° are degenerate in energy. Can the direction of stripes in Fig. 2(c) be adjusted by some means?
3. What is the difference between the proposed OAI in this manuscript and the recently reported boundary obstructed topological insulator TIGaTe₂ (Adv. Sci. 2022, 9, 2202564), which also has obstruction of Wannier charge center?
4. For the experimental side, I wonder how much one could rely on the second derivative procedure to reveal bands that are vague in the raw data, i.e., the comparison between Fig. (a-f) and Fig. (g-l). Is this a generally accepted analytical method of ARPES data?

Reviewer #2 (Remarks to the Author):

In this manuscript Liu et al. utilizes STM and ARPES measurements to investigate surface states of SrIn₂P₂ which is an obstructed atomic insulator candidate. This candidacy is supported by DFT calculations presented in this manuscript and its references which show existence of charge accumulation centers (virtual atom sites) between adjacent QLs. Associated bulk states cross the fermi level. When the sample is cleaved across the plane between adjacent QLs, these bulks states supposed to become surface states which can display half occupancy.

While the proposed surface states localized at virtual atom sites are likely to present very interesting physics, the surface states observed by the authors are fundamentally different as the surface goes through a reconstruction and resemble ordinary surface states produced

by dimerized dangling bonds rather than a novel bulk-boundary correspondence. Regardless of this outcome the results presented in the manuscript is still interesting and significant. Conclusions are well supported with sound methodology and data analysis. Therefore, I think that the manuscript can be published in Nature Communications. However, the authors should better clarify the difference between the surface states they observe and ordinary surface states of say Si(100) or other QL components.

Such as?

Is the charge density on the surface significantly higher?

Is the surface states resilient to contamination or disorder ? (judging from Fig S4 and S5 the answer seem to be no. But authors may have more data)

Apart from the above question manuscript is well written. However, I suggest that the authors revisit the following items to improve the quality of manuscript

In page 5, paragraph 2: the phrase “ maps show a much stronger signal within $V_B = 0.9$ to 1.2 V,” is very vague. They should clarify this observation perhaps by discussing the contrast inversion at defect sites of the dI/dV maps or putting a scale bar for the figures.

It is hard to follow BS2 bulk bands in Fig. 3. The authors can try to make red lines thinner or put the figures without lines in SI.

The measured height difference in STM images is just 5pm while the DFT calculations predict 113 pm. While some discrepancy is expected ~ 20 times difference seem a lot. The authors can answer this question by providing a simulated STM images (and line profile). This will also illuminate if the bright stripes are topology related (bright line at raised atom sites) or charge density related (bright line at lower atom sites)

The authors claim that they observe a “highly localized in-gap state with narrow bandwidth above the Fermi level” however the STS data show broad (~ 400 mV) peak at conduction band edge.

The sample temperature during the STM measurements should be provided.

Typos:

Figure S6 caption: I believe the $V_b = -700$ mV not “-700 V”

Reviewer #3 (Remarks to the Author):

In this manuscript, Professor Liu and colleagues studied structural and electronic properties of SrIn₂P₂ and concluded that this material could host obstructed surface states. However, I do not find the correspondence for most of the claims in the experimental or computational data. Therefore, I do not recommend this article for publication. Here are my main concerns and comments.

1)How did Authors conclude that their data shows emergent surface states by the bulk-boundary correspondence? Neither experimental nor computational data shows that the band structure holds this theory. This term is used for evaluation of the Z_2 invariant number across an interface between the topological insulator and normal insulator. It enforces bulk band gap closing keeping the system metallic no matter where the Fermi level sits. This is fundamental mechanism for topological materials to have edge or surface states when they are in contact with a normal insulator. This is not the case for the electronic structure of SrIn₂P₂. Predicted surface states do not connect the conduction and valence bands so that it can be a normal insulator. This is discussed in Ref. 18 of the manuscript. Indeed, this is only reference that really talks about the obstructed surface states. But it is not quite the same one as discussed in the manuscript.

2)In the manuscript, lines 4-5 of page 3 state “In such reconstructed areas an unusual, highly localized in-gap state with narrow bandwidth above the Fermi level is observed.” This is based on Fig. S4. How can Authors make sure it is not a bulk band? When one considers the calculated band structure given in Figure 4c, bulk band is the only one located at 1 eV

after the Fermi level is shifted 0.25 eV as suggested by Authors.

3) How did Authors conclude that surface states have Kramer's degeneracy at the time-reversal invariant momenta points? Visually they are overlapped over a wide range of k-points (lines 25-26 of page 3). Based on this statement, the surface states after split should also hold Kramer's degeneracy. I do not see why not. This means they are 8-fold degenerate band structure at M and K points before split. Is this correct? Otherwise, how is Kramer's degeneracy lifted or can it be 8-fold degenerate?

4) In line 29, "metallic obstructed surface states" is a wrong term. System can be turned into the normal insulator by moving the Fermi level across the bulk band gap (Rev. Mod. Phys. 82, 3045 (2010)). Meanwhile SrIn₂P₂ is already called semiconductor in the manuscript.

5) Why the lower branch of the surface states is absent in the STS spectra given in Fig. S3? I do not quite understand physical mechanism behind the logic to attribute negative differential conductance to a special type of surface state.

6) Why the constant energy cuts given in Fig S7 and S8 do not exhibit the any sign of surface states at M and K points in contrast to the computed band structure?

7) In Figure 4c, In atom is shifted 113pm (1.13 Angstrom). This is over 18% expansion compared to thickness of a QL that is almost equivalent to creating a vacancy. Very likely, that is why the charges accumulate only around the left In atoms. In this case, STS spectra and computed band structure are not equivalent as claimed. Otherwise, Authors should make this point clearer.

It sounds that the Authors indirectly claim that they observe a special type of topological surface states. I do not see any evidence for that. Neither bulk boundary correspondence nor metallic surface state exists. Even though the ARPES data provides some evidence for quasi-2D states, I do not see enough match between the computed and experimental band structure. Overall dispersions and binding energies of the computed one are way off compared to the experimental band structure for the proposed surface states. Therefore, I do not recommend the manuscript for publication.

Reply to Reviewer Comments

Reply to Reviewer #1 (Remarks to the Author):

In the present manuscript, Liu et al. utilize STM and ARPES measurements as well as DFT calculations to study the low-lying electronic structure of SrIn₂P₂, demonstrating a new type of surface state that originated from the so-called “obstructed surface states (OSS)”. Such surface states appear on obstructed atomic insulators, a novel class of materials distinguished from trivial atomic insulators with potential applicability on surface catalysis. The observed surface states differ from the theoretical OSSs in that they split in energy due to a surface structural reconstruction, rendering two branches of surface bands, one on each side of the Fermi level. This reconstruction eliminates the partial occupancy of electrons in the original OSS picture, yet preserves the spatial locality of the OSSs, which is key ingredient for catalytic activity.

I found that this manuscript provided solid experimental and computational evidence supporting the observation of this surface state. To my knowledge, it is the first report of spectroscopic realization of such previously unstudied low-dimensional electronic structure. The STM and ARPES data is systematic and of high quality, sufficient to rule out other possibilities of the observed phenomena. The DFT calculations also provided a rational scenario to explain the experimental observations. Based on these points, I tend to recommend this paper to be published in Nature Communications, provided the following comments are addressed appropriately.

Author Response:

We thank the Referee for the tendency to recommend the publication of our manuscript in *Nature Communications*. In particular, the Referee finds that our paper “provided solid experimental and computational evidence supporting the observation of this surface state”, and is “the first report of spectroscopic realization of such previously unstudied low-dimensional electronic structure”. Indeed, we uncovered by spectroscopic means an unforeseen type of surface state in a candidate obstructed atomic insulator, which may have potential applications in photocatalysis. In the following paragraphs, we address the Referee’s concerns in an itemized manner.

1. I wonder the difference between the surface band structure of OWCC termination and other terminations that can exist in SrIn₂P₂ (such as breaking Sr-P or In-P bond). For the OAI system, will they carry special surface states when the termination does not cross the OWCC?

Author Response:

We thank the Referee for his/her important question on possible surface states of SrIn₂P₂. In principle, obstructed Wannier charge centers (OWCCs) are restricted by the crystal symmetry at empty Wyckoff positions away from the real atoms, which could lead to obstructed surface states (OSSs) pinned at the Fermi level on the boundary due to filling anomaly. Such filling anomaly is caused by the mismatch between the number of electrons required to satisfy charge neutrality and the crystal symmetry when the boundary crosses the OWCC. Therefore, when the boundary does not cross the OWCC, filling anomaly will not occur, and only ordinary surface states will be expected due to the different chemical environment of the surface.

In fact, we performed DFT calculations on the surface states of other possible (0001) terminations of SrIn_2P_2 which do not cut through the OWCCs, including the terminations that break the In-P bonds and the Sr-P bonds [Fig. R1(b), (d)]. The surface states on all of these terminations show dispersion distinct from the one that breaks the In-In bonds (the one that cuts through the OWCCs), but none of them passes the Fermi level. In these cases, the whole system (bulk + surface) is insulating. These surface states are ordinary surface states, not obstructed surface states.

To clarify this issue, we added a new section named “Surface states on different (0001) terminations of SrIn_2P_2 ” in the revised Supporting Information.

Fig. R1. Surface states formed on different (0001) terminations of SrIn_2P_2 (also shown in the new Fig. S15 in the revised Supporting Information). (a), (c) Schematic illustration of SrIn_2P_2 (0001) termination that breaks the (a) In-P and (c) Sr-P bonds. (b), (d) DFT-calculated surface states on terminations that break (b) the In-P bonds and (d) the Sr-P bonds, respectively, using a SrIn_2P_2 slab model. The blue bands represent the projected surface states formed by Terminations 1 and 3, while the red bands represent the projected surface states formed by Terminations 2 and 4. The gray bands represent the bulk bands.

2. For the surface reconstruction supported by DFT calculation, the stripes along three directions with an angle of 120° are degenerate in energy. Can the direction of stripes in Fig. 2(c) be adjusted by some means?

Author Response:

We thank the Referee for this question. Domains of several micrometers with stripes along all three

directions can be observed on the sample surface, indicating their energy degeneracy. We can hardly modify the orientation of the stripes within the domain once it is cleaved. However, near the domain boundaries, the disordered stripes may response to a large bias voltage. Fig. R2 shows two topographic images obtained before and after an STS measurement with a large voltage range on the same field of view, where the direction of a particular stripe (marked by the dotted white lines) is modified because of the application of a large bias.

To address this question, we added the above discussion in the revised main text and added Fig. R2 as a new graph Fig. S3 in the revised Supporting Information. On page 4 of the revised main text, it reads “Domains of several micrometers with stripes along all three directions with an angle of 120° can be observed.”

Fig. R2 (also shown in Fig. S3 in revised Supporting Information) Direction of the stripes near defects can be modified by a large bias voltage. (a,b) Two topographic images ($V_B = -1$ V) obtained on the same field of view before (a) and after (b) STS scan with a large-bias-voltage (-2.6 V). The dotted white lines marks the modified stripes.

3. What is the difference between the proposed OAI in this manuscript and the recently reported boundary obstructed topological insulator TlGaTe₂ (Adv. Sci. 2022, 9, 2202564), which also has obstruction of Wannier charge center?

Author Response:

We have also noticed this article by N. Mao *et al.* They proposed an orbital-shift-induced boundary obstructed topological insulators (BOTIs) with fragile topology, and named it the “boundary obstructed fragile TIs” (BOFTIs). In their work, only under certain *open boundary conditions*, some Wannier orbitals may leave the original Wyckoff position and move to another occupied Wyckoff position on the boundary. Such boundary obstruction will yield a surface state dependent on the boundary polarization direction, which is definitely a boundary effect. In their discussions on the surface states of TlGaTe₂, only specified terminations with charge transfer from the Tl-*p* orbital to the Ga-*p* orbital can yield a metallic surface state. However, for OAI candidates, OWCCs are obstructed on the empty positions by crystal symmetry, which is a bulk topological phase distinct from its corresponding atomic limit. Therefore, symmetry indicators and real-space

invariants would give trivial topological phases for BOFTIs but nontrivial topological phases for OAIIs. The difference in topology is the most important distinction between the two types of materials.

4. For the experimental side, I wonder how much one could rely on the second derivative procedure to reveal bands that are vague in the raw data, i.e., the comparison between Fig. 3(a-f) and Fig. 3(g-l). Is this a generally accepted analytical method of ARPES data?

Author Response:

We thank the Referee for his/her concern on the details of this experimental technique. The second order derivative and curvature procedure applied in our manuscript is a technique used for highlighting the band dispersion relation in the raw ARPES images. This method can help tracking the exact k - E locations of the bands by increasing the sharpness of the dispersion relation. The reliability and efficiency of this method were checked by the authors of Ref. 38 on both the experimental data and the simulated data. The code used in our data processing also comes from Ref. 38. In Figs. 3(g)-(l), we used the second order curvature method to highlight the bulk band dispersion and the residual intensity outside the $\bar{\Gamma}$ hole pocket, which is attributed to the surface state. In ARPES studies, it is a generally accepted analytical method applied widely in many published articles. Here we list several recent articles which utilize this method to analyze the ARPES data [1-4].

Reply to Reviewer #2 (Remarks to the Author):

In this manuscript Liu et al. utilizes STM and ARPES measurements to investigate surface states of SrIn₂P₂ which is an obstructed atomic insulator candidate. This candidacy is supported by DFT calculations presented in this manuscript and its references which show existence of charge accumulation centers (virtual atom sites) between adjacent QLs. Associated bulk states cross the fermi level. When the sample is cleaved across the plane between adjacent QLs, these bulks states supposed to become surface states which can display half occupancy.

While the proposed surface states localized at virtual atom sites are likely to present very interesting physics, the surface states observed by the authors are fundamentally different as the surface goes through a reconstruction and resemble ordinary surface states produced by dimerized dangling bonds rather than a novel bulk-boundary correspondence.

Regardless of this outcome the results presented in the manuscript is still interesting and significant. Conclusions are well supported with sound methodology and data analysis. Therefore, I think that the manuscript can be published in Nature Communications.

Author Response:

We thank the Referee for directly recommending the publication of our manuscript in *Nature Communications* based on the scientific significance, sound methodology and reliable data analysis. Although the observed surface states differ from the original picture of the obstructed surface states because of the surface reconstruction, the “reconstructed” states still exhibit novel characteristics that are derived from the obstructed surface states (OSS). From Fig. 4 in the main text, we notice that the charge centers on the reconstructed surface of SrIn₂P₂ are highly localized across the surface, and do not coincide with the top In atoms. Instead, they are protruding out of the surface into the vacuum above, resembling the case of a pristine OSS where the obstructed Wannier charge centers (OWCCs) are situated at empty sites that float above the surface. In STM measurements, such unusual surface charge distribution gives rise to a pronounced STS peak followed by a negative differential conductance (NDC) that extends across a significant spatial area on the ordered surface. Such punctiform array of spatially localized surface states provides huge number of active sites for atoms/molecules adsorption and electron transfer, making the cleavage surface on this type of materials a promising candidate for photocatalysis. In the following paragraphs, we address the Referee’s concerns in an itemized manner.

However, the authors should better clarify the difference between the surface states they observe and ordinary surface states of say Si(100) or other QL components.

Such as?

1 Is the charge density on the surface significantly higher?

2 Is the surface states resilient to contamination or disorder ? (judging from Fig S4 and S5 the answer seem to be no. But authors may have more data)

Author Response:

We thank the Referee for this constructive question. As noted by the Referee, the surface state we found in SrIn₂P₂ does share features similar to an ordinary surface state, such as a high surface charge density and volatility against contamination or disorder (as shown in Fig. R3, a point defect is enough to destroy the state).

Fig. R3. Another set of data showing how a point defect destroys the STS peak and the following negative differential conductance (also shown in new Fig. S6 in the revised Supporting Information). (a) STM topography scanned on an ordered $\sqrt{3}a \times 1a$ reconstructed region other than that shown in Fig. 2 of the main text ($V_B = 1.3$ V, $I_t = 500$ pA). (b) STS line cuts along the blue arrow in (a). Both the strong peak centered at $V_B = 0.7$ V and the negative differential conductance that follows at higher biases disappear near a point defect of the surface.

However, what truly sets our surface state apart is one significant difference that can hardly be ignored: the presence of a persistent negative differential conductance (NDC) following the strong STS peak all over the defect-free area on the crystal surface. In contrast, ordinary surface states of Si(100) or other QL components shows high charge density peaks without corresponding NDC.

Negative differential conductance (NDC) is an uncommon property where an increase in bias voltage results in a decrease in current. It occurs when a resonant tunneling structure is present, where electrons can tunnel through potential barriers at certain energy levels with complete transmission. NDC is a key characteristic of resonant tunneling.

In transport measurements, such a phenomenon is usually observed in resonant tunneling diodes made up of various heterostructures, as the presence of spatially separated energy levels is a prerequisite. Various heterostructures have been observed with NDC as a result of resonant tunneling, ranging from quantum well structures of semiconductors (AlAs/InGaAs, Si/SiGe, etc.) to Dirac fermions with a thin dielectric spacer (graphene/h-BN/graphene) [5].

In STM measurements, NDC can provide valuable insights into the underlying resonant tunneling structure between the tip and sample surface. The observations of NDC following a strong STS peak in a single crystal surface are particularly rare, indicating both a high local density of state (LDOS) and a resonant tunneling structure in the tunneling geometry near the high LDOS energy level. To the best of our knowledge, such observations have been only reported in the following cases:

(1) adatoms/molecules that are sparsely distributed on the sample surface, forming a metal-adatom/molecule-metal junction, so resonant tunneling between the tip and the adatom/molecule orbitals gives rise to the STS peak and the associated NDC [6, 7]; When more than one molecule is involved in the tunneling process (a chain of molecules), NDC could be also a result of the resonant tunnelings from multiple molecular orbitals.

(2) two-dimensional materials where surface states with both narrow bandwidth and spatial localization could be available. Such as resonant tunneling through

- a) layer-polarized van Hove singularity in bilayer graphene [8].
- b) magnetic field-induced edge states at the npn junction boundaries in graphene [9].
- c) the sublattice and layer localization of the nearly flat bands in ABC trilayer graphene [10].

For all the cases above, the NDC-following-STs-peak behavior is not global. It occurs only at atomic sites where a resonant tunneling structure exists: adatom/molecule sites with orbital states in Case 1; the B_T atomic site where the layer-polarized van Hove singularity can be probed in Case 2a; the boundary of npn junction where the Landau levels reside in Case 2b; and the A_1 atomic site with the sublattice and layer-localized nearly flat band in ABC trilayer in Case 2c.

In our case, all the defect-free area on $SrIn_2P_2$ shows similar NDC following strong STS peak behavior, indicating a particularly unusual origin. It is worth noting that near the STS peak energy, **dramatic height steps (~ 300 pm) appear abruptly at the originally flat region between areas with slightly different doping**. As shown in Fig. R4, the field of view shows a relatively flat surface at the defect-free area [Fig. R4(c)] at bias voltage $V_B = 1.6$ V. The STS peak positions at the defect-free area are plotted in Fig. R4(a), indicating a slight doping difference. When we obtain the topographic image at $V_B = 0.9$ V on the same area [Fig. R4(b)], those regions with the same

Fig. R4 (also shown as the new Fig. S10 of the revised Supporting Information) Energy distribution of the STS peak induces large difference of the apparent height in the topographic image. (a) A map showing the energy position of the STS peaks of the defect-free area, indicating the doping difference of the sample surface. Data is extracted from Fig. S8 by fitting the dI/dV vs. V plot with Gaussian equations in each pixel. (b) and (c) Topographic images of the same field of view at (b) $V_B = 0.9$ V, $I_t = 500$ pA, and (c) $V_B = 1.6$ V, $I_t = 500$ pA showing an abrupt rise in elevation ($\Delta z \approx 300$ pm) of areas with slight doping difference.

STS peak energy (0.9 V) experience a sudden uplift of ~ 300 pm, as if there appears a floating cloud.

The STM topographic image is a combination of surface corrugation and surface electronic state [11]. Such an abrupt height change near the STS peak energy marks a sudden appearance of an extremely high level of surface charge of density. Although it is difficult to quantitatively evaluate experimentally how much larger the LDOS peak is compared to other conventional surfaces, we are confident that the manifestation of NDC provides valuable evidence for the presence of a ubiquitous structure for resonant tunneling beneath the STM tip – a surface state that is likely localized both in energy and in space along the z direction. This is consistent with our DFT calculation, that the surface state originates from the OWCCs where no atom resides. The electron cloud of these obstructed charge densities is much closer to vacuum than real atoms, making it the most possible omnipresent structure for resonant tunneling in our study.

Though calculations have shown that In-In bonds are the most easily broken bond on the (0001) surfaces of SrIn_2P_2 , we have also performed additional DFT calculations on the surface that breaks chemical bonds other than the In-In bond. Our calculation shows no OWCC on those surfaces, so at least the charge density is expected to be centered at the surface atoms rather than the OWCCs between atoms. In Fig. R5 we present the DFT calculations of the partial charge densities on an exemplified cleavage surface without OWCCs (the one that breaks the In-P bonds) and compare it with that on the surface with OWCCs. Compared with the ordinary termination without OWCCs [Fig. R5(b)], the surface charge of the termination crossing the OWCCs has significant deviation from the surface atoms [Fig. R5(a)]. Importantly, such deviation is found to be retained after surface

Fig. R5. DFT calculated partial charge density at different terminations (also shown as Fig. S19 of the revised Supporting Information). (a) The termination that cross the OWCCs. (b) The termination with surface P atoms, not crossing the OWCCs. The golden regions represent the partial charge density distributions evaluated in the energy range of $-0.5 - 0.5$ eV comparing to the Fermi level.

reconstruction (Fig. 4 in our main text).

In summary, we believe that the “NDC following STS peak” observation provides valuable insight into an omnipresent structure for resonant tunneling on SrIn₂P₂ surface, marking the key difference comparing OWCC-originated surface states and ordinary high charge density peaks.

To address this issue, we have added the discussion of the importance of NDC in the main text.

- 1) We added the phrase “followed by negative differential conductance” to the sentence “The upper branch is marked with a striking differential conductance peak followed by negative differential conductance,” in the abstract.
- 2) we added “signified by the presence of a persistent negative differential conductance (NDC) following a strong STS peak all over the defect-free area on the crystal surface” in Paragraph 1, Page 3 of the main text.
- 3) We replaced “localized state above the Fermi level (upper branch).” with “localized state following with unusual negative differential conductance above the Fermi level (upper branch).” in Paragraph 1, Page 3 of the main text.
- 4) We added “It should be highlighted here that the observation of negative differential conductance following a strong STS peak in a single crystal surface is particularly rare, indicating both a high local density of state (LDOS) and a resonant tunneling structure in the tunneling geometry near the high LDOS energy level.” in Paragraph 1, Page 5 of the main text.
- 5) We replaced “These spectroscopic experiments resolve an extraordinary localized surface state above the Fermi level,” with “These spectroscopic experiments resolve an extraordinary localized surface state with rarely reported negative differential conductance above the Fermi level,” in Paragraph 1, Page 9 of the main text.

We have also added two new sections named “Peak energy distribution and apparent height in STM topography” and “Calculated partial charge density on different (0001) terminations of SrIn₂P₂” in the revised Supporting Information.

3 Apart from the above question manuscript is well written. However, I suggest that the authors revisit the following items to improve the quality of manuscript

3.1 In page 5, paragraph 2: the phrase “maps show a much stronger signal within $V_B = 0.9$ to 1.2 V,” is very vague. They should clarify this observation perhaps by discussing the contrast inversion at defect sites of the dI/dV maps or putting a scale bar for the figures.

Author Response:

We thank the Referee for carefully reading our manuscript and providing the helpful suggestions. We have added the following sentence discussing the contrast inversion at the defect sites in the main text: “The dI/dV maps show a much stronger signal within $V_B = 0.9$ to 1.2 V, which agrees with the pronounced peak seen on the STS spectra. The rapid filling of the observed surface state

within these biases likely leads to the contrast inversion at the defect sites in the dI/dV maps.” We have also added scale bars for the dI/dV maps in the revised version [Fig. 2(f) in the main text, Figs. S8 and S9 in the Supporting Information].

3.2 It is hard to follow BS2 bulk bands in Fig. 3. The authors can try to make red lines thinner or put the figures without lines in SI.

Author Response:

We thank the Referee for the advice. In Fig. R6 (Fig. S14 in the revised Supporting Information) we replot Fig. 3(g)-(l) in the main text without the red lines that represent the DFT calculated data.

Fig. R6 (also shown as Fig. S14 of the revised Supporting Information). Second order curvature analysis of band structure of SrIn₂P₂ at different k_z s. (a)-(f) Second order curvature analysis along energy distribution curves for spectra in Fig. 3 (a)-(f). Here we removed the red lines that represent the DFT calculated bands to make BS1 and BS2 clearer. White arrows: dispersion of the OSS-derived surface state (SS) and the bulk bands BS1 and BS2; green arrows: tail of the SS outside the hole-pocket region.

3.3 The measured height difference in STM images is just 5pm while the DFT calculations predict 113 pm. While some discrepancy is expected ~ 20 times difference seem a lot. The authors can answer this question by providing a simulated STM images (and line profile). This will also illuminate if the bright stripes are topology related (bright line at raised atom sites) or charge density related (bright line at lower atom sites)

Author Response:

We thank the Referee for the insightful comment. The measured height difference for the stripe-like pattern in Fig. 2(c) is about 5 pm at $V_B = 1.6$ V. This height difference varies at different biases but mostly stays within 10 pm. It is indeed one order of magnitude smaller than the theoretically predicted reconstruction.

As noted by the Referee, the measured height is both topology-related and charge density related. In our case, charge density plays a significant role. For example, in Fig. R4 in the previous discussion, we have observed about ~ 300 pm apparent height change at the STS peak energy as a result of surface charge density change rather than the locations of surface atoms. Our DFT calculation has shown that surface reconstruction will give rise to the splitting of the pristine obstructed surface state, leading to a charge distribution of the upper branch of the surface state

Fig. R7. Hint from DFT calculation on why the STM-observed height difference between neighboring In atoms is much smaller than the DFT-calculated value (also shown as the new Fig. S20 of the revised Supporting Information). The figure shows an exemplified charge density contour (golden bubble) surrounding the lower In atom, as well as the actual In atom (pink ball) on the right. What an STM topographic image observed is roughly the height difference between the left contour and the right atom (marked as “STM”), while the DFT-determined height difference between atoms is the actual height difference between the two pink balls, marked as “DFT”. The two values are unrelated, and need not be comparable in values.

concentrated on the In atom with a lower reconstruction position (the left atom), as shown in Fig. 4(c). More importantly, such obstructed charge densities are predicted to be constrained towards the empty sites far away from the real atoms, effectively lifting the measured height of the In atom reconstructed at a lower position compared with its neighboring In atoms.

Fig. R7 shows an exemplified charge density contour for a certain charge density value, where the effective radius of the charge density distribution for the lower In atom goes ~ 20 pm higher than the upper In atom. Please note that the effective radius of the charge density distribution can be modified for a different cut-off value, which has an experimental correlation to the tunneling parameters, leading to varying measured height difference.

In light of the above arguments, we believe that the measured height difference in STM images is a combined effect of atomic site shift and the resulting obstructed surface states. To address the Referee’s comment, we have added the above discussion in the newly-added section “Peak energy distribution and apparent height in STM topography” in the revised Supporting Information.

3.4 The authors claim that they observe a “highly localized in-gap state with narrow bandwidth above the Fermi level” however the STS data show broad (~ 400 mV) peak at conduction band edge.

Author Response:

We thank the Referee for bringing this up. The valence electrons of SrIn_2P_2 are from *s* or *p* orbitals. For these electrons, a 400-meV bandwidth is relatively narrow. To make it more accurate, we use the term “relatively narrow” instead of “narrow” (Paragraph 1, Page 3 in the main text) and “surprisingly narrow” (Paragraph 3, Page 6 in the main text) in the revised manuscript.

3.5 The sample temperature during the STM measurements should be provided.

Author Response:

We thank the Referee for pointing out this. We added the following sentence in the **Methods** section in our revised manuscript: “All the STM measurements in this manuscript were done at 5 K.”

4 Typos:

Figure S6 caption: I believe the $V_b = -700 \text{ mV}$ not “-700 V”

Author Response:

We thank the Referee for carefully reading the manuscript and the typo correction. It is now fixed in our revised manuscript.

Reply to Reviewer #3 (Remarks to the Author):

In this manuscript, Professor Liu and colleagues studied structural and electronic properties of SrIn₂P₂ and concluded that this material could host obstructed surface states. However, I do not find the correspondence for most of the claims in the experimental or computational data. Therefore, I do not recommend this article for publication. Here are my main concerns and comments.

Author Response:

We thank the Referee for the reading of our paper. In the following paragraphs, we present an itemized response to each of the Referee's concerns. We hope that the Referee will find that our claims are in fact supported by our experimental and computational data, and the usage of terminologies in the paper is logically justifiable and trackable in the literature.

1) How did Authors conclude that their data shows emergent surface states by the bulk-boundary correspondence? Neither experimental nor computational data shows that the band structure holds this theory. This term is used for evaluation of the Z_2 invariant number across an interface between the topological insulator and normal insulator. It enforces bulk band gap closing keeping the system metallic no matter where the Fermi level sits. This is fundamental mechanism for topological materials to have edge or surface states when they are in contact with a normal insulator. This is not the case for the electronic structure of SrIn₂P₂. Predicted surface states do not connect the conduction and valence bands so that it can be a normal insulator. This is discussed in Ref. 18 of the manuscript. Indeed, this is only reference that really talks about the obstructed surface states. But it is not quite the same one as discussed in the manuscript.

Author Response:

We thank the Referee for the comment. It seems that the main discrepancy here is the understanding and definition of the terminology, i.e., bulk-boundary correspondence. While we agree with that this term is widely used “for evaluation of the Z_2 invariant number across an interface between the topological insulator and normal insulator”, we also feel that the connotation of this term is broader than Z_2 topological insulator, and even condensed matter systems. In the following we list some concrete examples in the literature.

Bulk-boundary correspondence does not limit to topological systems, but exists in various fields in physics. As a nice example, in Nomura and Nagaosa, “Surface-Quantized Anomalous Hall Current and the Magnetoelectric Effect in Magnetically Disordered Topological Insulators”, Phys. Rev. Lett. **106**, 166802 (2011), it is written that “*Bulk-surface correspondence has an essential role in a large variety of phenomena in condensed matter physics, such as ferroelectricity, diamagnetism, the Meissner effect, and the quantum Hall effect.*” Therefore, bulk-boundary correspondence is a concept that appeared frequently in classical electromagnetism. As a further example, Ramazanoglu *et al.*, “Bulk-boundary correspondence in soft matter”, Phys. Rev. E **100**, 020702 (2019) stated that, “The connection between a material's bulk and its boundary has been one of the guiding principles in **several branches of physics** in the last decade. The main idea is that the boundary of the system would feature excitations that do not occur in the bulk, yet the physics on the boundary is still determined by the properties of the bulk. ... **The holographic principle in high energy physics, also known as gauge-gravity duality, is another example of**

the *bulk-boundary correspondence* where the spectrum of the strongly interacting gauge theory in four space-time dimensions is connected to the weakly interacting theory on the three-dimensional boundary via duality.”

Even in the field of topological systems, *bulk-boundary correspondence* can be used for interfaces between various topologically non-equivalent phases (not limited to Z_2 topological insulators). In fact, this term has been widely used for description of 1D topological insulators and higher order topological insulators. Examples include: (i) 1D Zak phases. In Rhim *et al.*, “Bulk-boundary correspondence from the intercellular Zak phase”, Phys. Rev. B **95**, 035421 (2017), it is mentioned that “In this case, the conventional *bulk-boundary correspondence* states that there are boundary modes if the Zak phase is nontrivial, $\gamma = \pi$, while $\gamma = 0$ is considered a trivial insulator without surface modes.” (ii) High-order TI. In Schindler *et al.*, “Fractional corner charges in spin-orbit coupled crystals”, Phys. Rev. Research **1**, 033074 (2019), the authors stated that “the topological *bulk-boundary correspondence* is extended to include **higher-order topological insulators (HOTIs)** which exhibit hinge or corner modes in 3D and corner modes in 2D.” (iii) Fragile TI. Song *et al.* discussed in Ref. 18 the theory of fragile topology, which exhibits a new type of bulk-boundary correspondence with gapless edges under “twisted” boundary conditions. The relation between such fragile topological phase and obstructed atomic insulator is discussed in Ref. 18, stating clearly that these two phases are not identical. (iv) In Chen *et al.*, “Bulk-boundary correspondence in (3+1)-dimensional topological phases”, Phys. Rev. B **94**, 045113 (2016), it is further clarified that, “The *bulk-boundary correspondence* is one of the most salient features of topologically ordered phases of matter. In topologically ordered states in (2+1) dimensions $[(2+1)d]$, **all essential topological properties in their bulk can be derived and understood from their edge theories**, such as quantized transport properties, properties of bulk quasiparticles (**fractional charge and braiding statistics thereof**), and the **topological entanglement entropy**, etc.” Note that the above examples are not necessarily related to the existence of an energy gap.

Overall, in our humble opinion, rather than a specific field in topological matter, the terminology of “*bulk-boundary correspondence*” reflects one of the frontiers of the research in physics. Historically, a new type of bulk-boundary correspondence can often inspire a wave of study both theoretically and experimentally, such as high-order TI, fragile topology, and braiding statistics. In our work, we strongly believe that the OAI phase also manifests a new type of bulk-boundary correspondence, which is recently predicted by various famous groups (Refs. 17, 21, and 22). Despite that OAIs share the trivial Z_2 number as that of the vacuum, they are a class of novel materials that could be considered “topological” because the bulk of such materials is different from that of the vacuum in that they possess different real-space topological indexes. As a result, the ideal surface states that appear at the boundaries, while disconnected with the conduction and valence bands, exhibit a metallic feature (filling anomaly induced obstructed surface state). Importantly, the occurrence of the surface states originates from the *bulk* electronic structure where the presence of charge centers on “empty sites” between real atoms (obstructed Wannier charge centers) is protected by topological invariants.

In our work, we discover an unforeseen type of surface state in a candidate obstructed atomic insulator material SrIn_2P_2 . Specifically, we uncover a pair of surface states in this material which is result of an adiabatic evolution from the half-filled obstructed surface states. The deviation from

the original obstructed surface states is found to be caused by a surface structural reconstruction. However, such evolution does not affect the nature of bulk electronic structures, as well as the *bulk-boundary correspondence*. The unique bulk-boundary correspondence demonstrates the robustness of such surface states, explaining its ubiquitous appearance on the intrinsic surface of SrIn₂P₂. Therefore, we believe that this work is of importance in the versatile fields of surface science, topological electronic structure and functional materials, and that it will have a high impact on the broad readership of *Nature Communications*.

2) In the manuscript, lines 4-5 of page 3 state “In such reconstructed areas an unusual, highly localized in-gap state with narrow bandwidth above the Fermi level is observed.” This is based on Fig. S4. How can Authors make sure it is not a bulk band? When one considers the calculated band structure given in Figure 4c, bulk band is the only one located at 1 eV after the Fermi level is shifted 0.25 eV as suggested by Authors.

Author Response:

First of all, the energy position of the STS peak varies at different sample locations. We have observed STS peaks located from 0.6 eV to 1.2 eV. Fig. R3 shows a region with a majority peak at ~ 0.7 eV. Fig. R4(a) shows another location with regions involving more changes, where the peak position tends to shift upwards/downwards near the different types of disorders (upwards in the circled green area and downwards in the squared pink area). Because our DFT calculation agrees with ARPES measurements, which is with less spatial resolution compared with STM, the coincidence noted by the Referee likely comes from a Fermi level shift that depends on the local doping level.

Fig. R3. Another set of data showing how a point defect destroys the STS peak and the following negative differential conductance (also shown in new Fig. S6 in the revised supplementary material). (a) STM topography scanned on an ordered $\sqrt{3}a \times 1a$ reconstructed region other than that shown in Fig. 2 of the main text ($V_B = 1.3$ V, $I_t = 500$ pA). (b) STS line cuts along the blue arrow in (a). Both the strong peak centered at $V_B = 0.7$ V and the negative differential conductance that follows at higher biases disappear near a point defect of the surface.

To clarify the issue, we have added the above discussion in a new section named “Peak energy distribution and apparent height in STM topography” in the revised Supplementary Material.

Secondly, we have observations that the peak is extremely sensitive to disorders: a) it disappears at the presence of a point defect [Fig. R3(b)]; b) the energy location of the peak is greatly affected when getting near the disorders near the surface [Fig. R4(a)]. Both are consistent with a surface origin rather than a bulk property.

Thirdly, as discussed previously in replying to Referee 2 on page 6, the appearance of omnipresent negative differential conductance (NDC) following STS peak marks a ubiquitous structure for resonant tunneling in the current path. To the best of our knowledge, a bulk origin can hardly give rise to the NDC feature with the proper geometry. The best picture describes our STM and ARPES observations consistently is the surface states originated from the obstructed Wannier charge centers.

Fig. R4 (also shown as the new Fig. S10 of the revised Supporting Information) Energy distribution of the STS peak induces large difference of the apparent height in the topographic image. (a) A map showing the energy position of the STS peaks of the defect-free area, indicating the doping difference of the sample surface. Data is extracted from Fig. S8 by fitting the dI/dV vs. V plot with Gaussian equations in each pixel. (b) and (c) Topographic images of the same field of view at (b) $V_B = 0.9$ V, $I_t = 500$ pA, and (c) $V_B = 1.6$ V, $I_t = 500$ pA showing an abrupt rise in elevation ($\Delta z \approx 300$ pm) of areas with slight doping difference.

3) How did Authors conclude that surface states have Kramer’s degeneracy at the time-reversal invariant momenta points? Visually they are overlapped over a wide range of k-points (lines 25-26 of page 3). Based on this statement, the surface states after split should also hold Kramer’s degeneracy. I do not see why not. This means they are 8-fold degenerate band structure at M and K points before split. Is this correct? Otherwise, how is Kramer’s degeneracy lifted or can it be 8-fold degenerate?

Author Response:

We thank the Referee for pointing out the issue of degeneracy of the surface states. For the non-magnetic material SrIn_2P_2 with termination, the time-reversal symmetry will be maintained on the surface, which protects 2-fold Kramer’s degeneracy at the time-reversal invariant momenta (TRIM) for all bands. In addition, the slab structure we employed has a thickness of 14 quintuple layers,

which is large enough to ignore the hybridization between two surfaces, so the surface bands will have an extra 2-fold degeneracy of the up and down surfaces connected by symmetry m_z . Therefore, the surface bands are 4-fold degenerate at the TRIMs (Γ and M points) and 2-fold degenerate at the non-TRIMs (K point) in the slab calculation of Fig. 1(e).

Furthermore, to discuss the surface reconstruction, we considered a $\sqrt{3}a \times 1a$ supercell as shown in Fig. 4 of the main text. As we discussed in Section XIII of the Supporting Information, the band structure of the supercell without surface reconstruction can be regarded as the direct folding of the band structure of the unit cell. Therefore, the folded boundary of the Brillouin zone will naturally be providing another 2-fold degeneracy of a band, leading to an 8-fold degeneracy of the surface bands at the TRIMs on the boundary, such as the M_2 point marked in Fig. S17(c) in the revised Supporting Information [Fig. R8(c) below].

In our figures of DFT calculated band structures, the visual overlap over a wide range of momenta is due to the small energy splitting of the bands and the projections of surface atoms (marked by yellow). To be more clear, we show our band structures without projection before and after Brillouin zone folding in Fig. S17 [Fig. R8]. It can be seen that Kramer's degeneracy only exists on the TRIMs. Upon the $\sqrt{3}a \times 1a$ surface reconstruction, time-reversal symmetry is not violated, so Kramer's degeneracy still survives.

We have modified Fig. S17 (Fig. R8) and added the following statement about the degeneracy of surface bands to the caption of Fig. 1 in the main text: “DFT-calculated bulk band structure of the In-terminated (0001) cleavage plane of $SrIn_2P_2$ with a 7-unit-cell slab model, in which the Γ and M points are 4-fold degenerate if the hybridization between two surfaces is ignored.”

Fig. R8 (also shown as Fig. S17 of the revised Supporting Information). The DFT calculated band structures of unit cell and $\sqrt{3}a \times 1a$ supercell without surface reconstruction. (a) Schematic illustration of the Brillouin zone folding. The solid hexagon is the Brillouin zone of the slab unit cell and the dashed rectangular is the Brillouin zone of the slab $\sqrt{3}a \times 1a$ supercell. (b), (c) Band structures of the slab unit cell [corresponding to Fig. 1(e) in the main text] and the slab $\sqrt{3}a \times 1a$ supercell [corresponding to Fig. 4(a) in the main text], respectively. The numbers in blue brackets represent the degeneracy of the surface bands at the high symmetry points that ignore the hybridization between two surfaces. The red dashed circles and the associated enlarged view (right) depict the eight-fold degenerate bands at the M_2 point on the boundary of the Brillouin zone after folding.

4) In line 29, “metallic obstructed surface states” is a wrong term. System can be turned into the normal insulator by moving the Fermi level across the bulk band gap (Rev. Mod. Phys. 82, 3045 (2010)). Meanwhile SrIn₂P₂ is already called semiconductor in the manuscript.

Author Response:

In our humble opinion, a condensed matter system is defined *metallic (conductive)*, if there exist energy bands that cross the *present* Fermi level. Even if a gap exists below and above the Fermi level, such that appropriate doping would shift the Fermi level into the gap, the present system *before* such Fermi level shifting is still defined as *metallic (conductive)*, since a significant value of electron *conductivity* is provided by the surface band. Similarly, we call transparent conducting oxides conducting because the Fermi level cuts some bands, even if it has a large gap (transparent) by shifting the Fermi level.

By saying “metallic obstructed surface states”, what we referred to is the theoretical obstructed surface state of SrIn₂P₂ (before the surface reconstruction), not the one we observed experimentally (after surface reconstruction). All the three times that “metallic obstructed surface states” appeared in the main text are listed below. The theoretically predicted obstructed surface state, no matter in the literature or our calculations before surface reconstruction, is indeed metallic. The metallicity of the ideal obstructed surface state is guaranteed by filling anomaly. This is in fact where the novelty of the obstructed surface state lies: once appear, it is guaranteed to cross the Fermi level.

Furthermore, **the term “metallic obstructed surface states” has also been used in the literature.** In Li *et al.*, Adv. Mater. **34**, 2201328 (2022), it is written in the abstract that “The obstructed Wannier charge centers (OWCCs) in OAI are pinned by symmetries at some empty Wyckoff positions so that surfaces that accommodate these sites are guaranteed to have *metallic obstructed surface states* (OSSs).” Therefore, the usage of this term is endorsed by B. A. Bernevig and C. Felser, the original theoretical proposers of obstructed atomic insulators themselves. Further, in Xu *et al.*, arXiv:2111.02433, it is stated that “it is possible to have a finite-size crystal with the OWCCs on the boundary and preserving the crystal symmetry, which exhibits the filling anomaly and hence gives rise to *metallic obstructed surface states* (OSSs) or hinge states.”

In the Referee Comment, “metallic” seems to require that the surface band cross the Fermi level *no matter where the Fermi level is* (i.e., the surface gap equals zero, which is the case of Z_2 topological insulator). But this is not what “metallic” means. In our opinion, an appropriate term describing this situation should be *gapless*, not *metallic*.

To avoid confusion, in the revised version we have used the term “***bulk semiconductor***” for SrIn₂P₂ to clarify its semiconducting nature in the bulk.

PS: In the following we list where the term “metallic obstructed surface states” appeared in the main text:

Line 29, Page 3: “*The half occupation of these states (filling anomaly) inevitably occurs if charge neutrality and certain crystalline symmetries (e.g., inversion symmetry of the slab) are*

simultaneously preserved [11-14], giving rise to metallic obstructed surface states.”

Line 17, Page 6: “Till now, we have gathered the following information about the new surface states observed in SrIn_2P_2 : (i) According to the theory of obstructed atomic insulators, a pair of relatively flat, metallic obstructed surface states should have appeared due to the filling anomaly for the undistorted surface. However, such half-filled bands are not observed both in our STM and ARPES measurements.”

Caption of Figure 1(c): “Illustration of the metallic obstructed surface states (OSSs) that originate from the OWCCs at the interface within the bulk band gap.”

5) Why the lower branch of the surface states is absent in the STS spectra given in Fig. S3? I do not quite understand physical mechanism behind the logic to attribute negative differential conductance to a special type of surface state.

Author Response:

We thank the Referee for this comment. Based on the calculation, most of the upper branch of the surface state is localized within the bulk gap. However, the lower branch is much more dispersive and buried within the bulk bands. Therefore, the signature STS peak as in the upper branch is not expected. As shown in Fig. 2(e) of the main text, the STS below the Fermi level shows a non-zero value, and no peak is observed. Several faint peaks can be distinguished in an STS with a smaller tunneling bias (Fig. R9), which are resulted from the combined density of states of the bulk and surface bands. Figuring out the exact band origin of these faint STS peaks is beyond the current experimental resolution.

For the NDC part, we have discussed NDC following an STS peak as a key feature distinguishes our surface state, as stated in the previous reply to Referee 2 on page 6. For the reviewer’s convenience, we copied the related discussion below:

Fig. R9. STS at small bias. The tunneling junction is set at $V_B = -0.7$ V, $I_t = 0.5$ nA.

Negative differential conductance (NDC) is an uncommon property where an increase in bias voltage results in a decrease in current. It occurs when a resonant tunneling structure is present, where electrons can tunnel through potential barriers at certain energy levels with complete transmission. NDC is a key characteristic of resonant tunneling.

In transport measurements, such a phenomenon is usually observed in resonant tunneling diodes made up of various heterostructures, as the presence of spatially separated energy levels is a prerequisite. Various heterostructures have been observed with NDC as a result of resonant tunneling, ranging from quantum well structures of semiconductors (AlAs/InGaAs, Si/SiGe, etc.) to Dirac fermions with a thin dielectric spacer (graphene/h-BN/graphene) [5].

In STM measurements, NDC can provide valuable insights into the underlying resonant tunneling structure between the tip and sample surface. The observations of NDC following a strong STS peak in a single crystal surface are particularly rare, indicating both a high local density of state (LDOS) and a resonant tunneling structure in the tunneling geometry near the high LDOS energy level. To our best knowledge, such observations have been only reported in the following cases:

(1) adatoms/molecules that are sparsely distributed on the sample surface, forming a metal-adatom/molecule-metal junction, so resonant tunneling between the tip and the adatom/molecule orbitals gives rise to the STS peak and the associated NDC [6, 7]; When more than one molecule is involved in the tunneling process (a chain of molecules), NDC could be also a result of the resonant tunneling from multiple molecular orbitals.

(2) two-dimensional materials where surface states with both narrow energy localization and spatial localization could be available. Such as resonant tunneling through

a) layer polarized van Hove singularity in bilayer graphene [8]

b) magnetic field-induced edge states at the npn junction boundaries in graphene [9]

c) the sublattice and layer localization of the nearly flat bands in ABC trilayer graphene [10]

For all the cases above, the NDC following STS peak behavior is not global. It occurs only at atomic sites where a resonant tunneling structure exists: adatom/molecule sites with orbital states in Case 1; the B_T atomic site where the layer polarized van Hove singularity can be probed in Case 2a; the boundary of npn junction where the Landau levels reside in Case 2b; and the A_1 atomic site with the sublattice and layer localized nearly flat band in ABC trilayer in Case 2c.

In our case, all the defect-free area on SrIn_2P_2 shows similar NDC following strong STS peak behavior, indicating a particularly unusual origin. It is worth noting that near the STS peak energy, **dramatic height steps (~ 300 pm) appear abruptly at the originally flat region between areas with slightly different doping.** As shown in Fig. R4, the field of view shows a relatively flat surface at the defect-free area [Fig. R4(b)] at bias voltage $V_B = 1.6$ V. The STS peak positions at the defect-free area are plotted in Fig. R4(a), indicating a slight doping difference. When we obtain the topographic image at $V_B = 0.9$ V on the same area, those regions with the same STS peak energy (0.9 V) experience a sudden uplift of ~ 300 pm, as if there appears a floating cloud.

Fig. R4 (also shown as the new Fig. S10 of the revised Supporting Information) Energy distribution of the STS peak induces large difference of the apparent height in the topographic image. (a) A map showing the energy position of the STS peaks of the defect-free area, indicating the doping difference of the sample surface. Data is extracted from Fig. S8 by fitting the dI/dV vs. V plot with Gaussian equations in each pixel. (b) and (c) Topographic images of the same field of view at (b) $V_B = 0.9$ V, $I_t = 500$ pA, and (c) $V_B = 1.6$ V, $I_t = 500$ pA showing an abrupt rise in elevation ($\Delta z \approx 300$ pm) of areas with slight doping difference.

The STM topographic image is a combination of surface corrugation and surface electronic state [11]. Such an abrupt height change near the STS peak energy marks a sudden appearance of an extremely high level of surface charge of density. Although it is difficult to quantitatively evaluate experimentally how much larger the LDOS peak is compared to other conventional surfaces, we are confident that the manifestation of NDC provides valuable evidence for the presence of a ubiquitous structure for resonant tunneling beneath the STM tip--- a surface state that is likely localized both in energy and in space along the z direction. This is consistent with our DFT calculation, that the surface state originates from the OWCCs where no atom resides. The electron cloud of these obstructed charge densities is much closer to vacuum than real atoms, making it the most possible omnipresent structure for resonant tunneling in our study.

Below are our additional arguments specified for this Referee Comment:

Fig. R10 describes the detailed process of resonant tunneling through the electron cloud originating from the OWCCs. The apex of the STM tip is a protruding atom (an atom that projects outward). Its spatial constraint will introduce localized electronic states like those in a quantum well. In ordinary conditions, no localized surface states reside outside the sample surface, and the tunneling current from the localized states on the tip contributes negligibly to the tunneling current compared with that from the tip's metallic states (states that come from the element of the tip itself, like W, Pt or Ir). Therefore, Bardeen's formula is valid:

$$\frac{dI}{dV} \propto \int_{-\infty}^{\infty} f'(E_F + \varepsilon) \rho_s(E_F - eV + \varepsilon) d\varepsilon \approx \rho_s(eV).$$

However, on SrIn_2P_2 , spatially-localized surface states situate higher than the sample's itinerant states in real space, deep into the vacuum region where no atoms reside, described as the obstructed

Fig. R10. Due to the localized states on the apex of the STM tip, filling of the localized surface states on the sample induces tunneling current anomaly and negative differential conductance (NDC) (also shown as Fig. S11 of the revised supplementary material). (a) Measured I vs. V curve on the surface of SrIn_2P_2 . The dotted black line is the presumed tunneling current without the localized-state resonance peak. (b) Illustration of tunneling current anomaly induced by local-state resonance. From left to right: When $V_B < V_{peak}$ (V_B being the bias voltage), the surface state peaks are not filled, no tunneling current anomaly is present. When $V_B \approx V_{peak}$, the surface state peaks are partially filled, tunneling current anomaly emerges. When $V_B > V_{peak}$, The surface state peaks are fully filled, there is again no tunneling current anomaly. The tunnelling current decreases with increasing bias voltage, giving rise to NDC. (c) Measured dI/dV versus V plot on the surface of SrIn_2P_2 . Due to the current anomaly induced by local-state resonance, the phenomenon of a peak following by a NDC appears.

Wannier charge center. The overlap between the protruding localized states on the sample and the protruding localized states on the tip strongly enhances the tunneling matrix element. Therefore, the part of the tunneling current from the tip's localized states can no longer be ignored. Instead, it dominates the measured tunneling current. The tunneling current can now be approximated as:

$$I \propto \int_0^{eV} \rho_s(E_F - eV + \varepsilon) \rho_T(E_F + \varepsilon) d\varepsilon.$$

Here the tip's density of states $\rho_T(E)$ is assumed to have a localized state near E_F [6, 11].

Without the tunneling channel from the apex of the tip, the I - V curve should be monotonic, as shown by the dotted black line in Fig. R10(a). When the two localized states line up for the resonant tunneling at $V_B \approx V_{peak}$, the measured I - V curve deviates from a monotonic one because of the dramatically enhanced tunneling. A current anomaly takes place. With such an anomaly in the I - V curve, a strong peak followed by an NDC appears in the dI/dV - V plot.

In summary, the NDC following a strong peak observed all over the defect-free area provides insightful hint for the obstructed electronic structure in SrIn₂P₂.

6) Why the constant energy cuts given in Fig S7 and S8 do not exhibit the any sign of surface states at M and K points in contrast to the computed band structure?

Author Response:

We thank the Referee for the comment on our ARPES data. The constant energy contours shown in Figs. S12 and S13 are integrated within an energy window of 50 meV centered around each binding energy. As shown in Fig. 3(a)-(e) and Fig. R12(a)-(b), the intensity of surface states near the \bar{K} and \bar{M} points is much weaker than the linear SS band and other bulk bands near the $\bar{\Gamma}$ point. Thus, the trace of surface state near the \bar{K} and \bar{M} points could hardly be observed in the constant energy contours. Instead, this residual intensity is easier to be observed in ordinary ARPES E - k cuts. By applying the second-order curvature procedure to the raw data, one could resolve the trace of the surface state near \bar{K} and \bar{M} [indicated by the green arrows in Fig. R11(a)-(f) and Fig. R12(c)-(d)], which qualitatively agree with the computed band structure.

To clarify this issue, we added Fig. S15 and S16 (same as Figs. R11 and R12) in the new section “ARPES E - k cuts along \bar{K} - $\bar{\Gamma}$ - \bar{K} and \bar{M} - $\bar{\Gamma}$ - \bar{M} ” of the revised Supporting Information.

Fig. R11 (Same as Fig. R6; also shown as Fig. S15 of our revised Supporting Information). Second order curvature analysis of band structure of SrIn₂P₂ at different k_z s. (a)-(f) Second order curvature analysis along energy distribution curves for the spectra in Fig. 3(a)-(f) of the main text. Here we remove the red lines that represent the DFT-calculated bands to make BS1 and BS2 clearer. White arrows: dispersion of the OSS-derived surface state (SS) and the bulk bands BS1 and BS2; green arrows: tail of the SS outside the hole-pocket region. **Note that the \bar{K} point is marked in (c) and (d), indicating that the residue intensity of the surface state extends to \bar{K} .**

Fig. R12 (also shown as Fig. S16 of our revised Supporting Information). ARPES E - k cuts at different photon energies along $\bar{M}\text{-}\bar{\Gamma}\text{-}\bar{M}$. (a)-(b) ARPES E - k cuts along $\bar{M}\text{-}\bar{\Gamma}\text{-}\bar{M}$ at $h\nu = 90$ and 76 eV, which correspond to $k_z = 4.741 \text{ \AA}^{-1}$ ($\sim 0.9 \pi/c$) and 5.114 \AA^{-1} ($\sim \pi/c$), here $c = 17.812 \text{ \AA}$. (c)-(d) Second order curvature analysis along the energy distribution curves of the spectra in (a)-(b). White arrows: dispersion of the OSS-derived surface state (SS) and the bulk bands BS1 and BS2; green arrows: tail of the SS outside the hole-pocket region. **Note that the \bar{M} point is marked in (c) and (d), indicating that the residue intensity of the surface state extends to \bar{M} .**

7) In Figure 4c, In atom is shifted 113pm (1.13 Angstrom). This is over 18% expansion compared to thickness of a QL that is almost equivalent to creating a vacancy. Very likely, that is why the charges accumulate only around the left In atoms. In this case, STS spectra and computed band structure are not equivalent as claimed. Otherwise, Authors should make this point clearer.

Author Response:

We thank the Referee for this comment.

1. A 18% shift of the upper In atom is not an outrageous value in surface science.

In the literature, we found many results in which the top atoms on the surface are elevated to a higher percentage than the 18%. For example, on the InP(110) 2×1 reconstructed surface, it is reported that the topmost In atom shifts for ~ 70 pm respecting to the topmost P atoms (Fig. R13(a) [12]). Considering the relatively small value of the vertical lattice constant of InP(110) (~ 200 pm), the In atom shift on InP(110) is about 35%, which is much more significant than what we proposed on SrIn₂P₂ (0001).

Fig. R13. Structural reconstruction of the InP(110) 2×1 and Si(111) 7×7 surface [11-13]. (a) Top view (upper) and side view (lower) of the 2×1 surface reconstruction of InP(110). Open circles are anions and hatched circles are cations. a_0 is the theoretical bulk lattice constant and $d_0 = \sqrt{2}a_0/4$. $\Delta_{1,\perp}$ is the displacement between the topmost In and P atoms, which is ~ 73 pm by calculation. (b) Top view (upper) and side view (lower) of 7×7 surface reconstruction of Si(111). The 7×7 unit cell is outlined by the red dotted line in the upper image. In each unit cell, there are 9 dimers (Open circle pairs), 12 adatoms (Heavily outlined circles), and a stacking-fault layer (Large open circles in the left and right triangle). Solid red circles are the rest atoms. Solid circles and dots represent atoms in the unreconstructed layers beneath the unreconstructed surface. At 4 corners of the 7×7 unit cell, there is a Si vacancy. In each 7×7 unit cell, there are 19 dangling bonds, of which 12 are from the adatoms, 6 from the rest atoms, and 1 from the corner hole vacancy.

2. The charges of the surface state not only accumulate around the left In atoms, but also around the right one, when both branches of the surface state are occupied.

The surface reconstruction leads to the splitting of the original obstructed surface states, where the charge density redistributes between the two branches of the split surface states. Fig. R14 shows the calculated results after reconstruction: the charges over the left In atom constitute the upper branch of the surface state [Fig. R14(b)], which is above the experimental Fermi level; while the charges over the right In atom constitute the lower branch of the surface state [Fig. R14(c)], which is below the experimental Fermi level. Compared with the upper branch, the lower branch is buried within the bulk states, so the tunneling does not necessarily give the same signature in STS signal as in the upper branch.

3. Even if we adopt the scenario that a vacancy appears on the surface, the omnipresent feature of NDC following the STS peak cannot be explained.

Among all the surface reconstructions, Si(111) 7×7 is the most famous one investigated by STM. In the classical dimer-adatom-stacking-fault (DAS) model shown in Fig. R13(b) [11,13], the 7×7 unit cell can be divided into an unfaulted triangle (left) and faulted triangle (right). Beyond the stacking-fault layer, 12 Si adatoms reside. On the corners of the 7×7 unit cell, a Si vacancy appears.

There are a total of 19 dangling bonds in the 7×7 unit cell, of which 12 are from the adatoms, 6 from the rest atoms, and 1 from the corner vacancy. Although there is a vacancy inside the corner hole, no NDC is observed on the Si(111) 7×7 reconstructed surface [6]. According to our previous discussion (Page 19 – 20), an OWCC-derived surface state is the most probable explanation for the omnipresent behavior of “an NDC following the STS peak”.

Fig. R14. The DFT calculated charge distributions of the split surface states. (a) The band structure of SrIn_2P_2 with $\sqrt{3} \times 1$ reconstructed surface. The orange bands represent the surface projected states. (b)-(c) The partial charge density distributions evaluated by the upper and lower branches of the split surface states with the k -points near the Y-M line.

References

- [1] Yu, T. L., Xu, M., Yang, W. T. *et al.* Strong band renormalization and emergent ferromagnetism induced by electron-antiferromagnetic-magnon coupling. *Nat. Commun.* **13**, 6560 (2022).
- [2] Ma, J., Nie, S., Gui, X. *et al.* Multiple mobile excitons manifested as sidebands in quasi-one-dimensional metallic TaSe_3 . *Nat. Mater.* **21**, 423–429 (2022).
- [3] Schröter, N.B.M., Pei, D., Vergniory, M.G. *et al.* Chiral topological semimetal with multifold band crossings and long Fermi arcs. *Nat. Phys.* **15**, 759–765 (2019).
- [4] Rao, Z., Li, H., Zhang, T. *et al.* Observation of unconventional chiral fermions with long Fermi arcs in CoSi . *Nature* **567**, 496–499 (2019).
- [5] Britnell, L., Gorbachev, R., Geim, A. *et al.* Resonant tunnelling and negative differential conductance in graphene transistors. *Nat Commun* **4**, 1794 (2013).
- [6] Lyo, I. W. & Avouris, A. Negative differential resistance on the atomic scale: implications for atomic scale devices. *Science* **245**, 1369 (1989).
- [7] Xue, Y.-Q., Datta, S., Hong, S.-H., *et al.* Negative differential resistance in the scanning-tunneling spectroscopy of organic molecules. *Phys. Rev. B* **59**, R7852(R) (1999).
- [8] Kim, K. S., Kim, T.-H., Walter, A. L., *et al.* Visualizing Atomic-Scale Negative Differential Resistance in Bilayer Graphene. *Phys. Rev. Lett.* **110**, 036804 (2013).
- [9] Li, S.-Y., Liu, H., Qiao, J.-B., Jiang, H., & He L. Magnetic-field-controlled negative differential conductance in scanning tunneling spectroscopy of graphene npn junction resonators. *Phys. Rev. B* **97**, 115442 (2018).
- [10] Yin, L.-J., Yang, L.-Z., Zhang, L., Wu, Q., Fu, X., Tong, L.-H., Yang, G., Tian, Y., Zhang, L.,

- & Qin, Z. Imaging of nearly flat band induced atomic-scale negative differential conductivity in ABC-stacked trilayer graphene. *Phys. Rev. B* **102**, 241403 (2020).
- [11] Chen, C. J. *Introduction to Scanning Tunneling Microscopy: Second Edition*. (Oxford University Press, 2007).
- [12] Alves, J. L. A., Hebenstreit, J., and Scheffler, M. Calculated atomic structures and electronic properties of GaP, InP, GaAs, and InAs (110) surfaces. *Phys. Rev. B* **44**, 6188 (1991).
- [13] Brommer, K. D., Needels, M., Larson, B., and Joannopoulos, J. D. Ab initio theory of the Si(111)-(7×7) surface reconstruction: A challenge for massively parallel computation. *Phys. Rev. Lett.* **68**, 1355 (1992).

REVIEWERS' COMMENTS

Reviewer #1 (Remarks to the Author):

After reading the reply, I think my concerns about this manuscript are well addressed and the manuscript has been revised accordingly. Hence, I recommend its publication.

Reviewer #2 (Remarks to the Author):

The revised manuscript by Liu et al. and the corresponding reply to reviewer comments fully address my concerns and questions about the manuscript. It also successfully answers the comments and questions the other two referees raised. Therefore, I recommend that this paper be published in Nature Communications.

Reply to Reviewer Comments

Reviewer #1 (Remarks to the Author):

After reading the reply, I think my concerns about this manuscript are well addressed and the manuscript has been revised accordingly. Hence, I recommend its publication.

Reviewer #2 (Remarks to the Author):

The revised manuscript by Liu et al. and the corresponding reply to reviewer comments fully address my concerns and questions about the manuscript. It also successfully answers the comments and questions the other two referees raised. Therefore, I recommend that this paper be published in *Nature Communications*.

Author Response:

To both reviewers: Thank you for the kind recommendation to publish this manuscript in *Nature Communications*. Your help in the reviewing process is highly appreciated.